# Electroencephalography Findings in Older Adults Undergoing Geriatric Treatment: A Surrogate for the Outcome?

**DOI:** 10.3390/brainsci12070839

**Published:** 2022-06-28

**Authors:** Marco Meyer, Stefanie Schmetsdorf, Thomas Stein, Ulrich Niemöller, Andreas Arnold, Patrick Schramm, Josef Rosenbauer, Karel Kostev, Christian Tanislav

**Affiliations:** 1Department of Geriatrics and Neurology, Diakonie Hospital Jung Stilling Siegen, 57080 Siegen, Germany; marco.meyer@diakonie-sw.de (M.M.); stefanie.schmetsdorf@diakonie-sw.de (S.S.); thomas.stein@diakonie-sw.de (T.S.); ulrich.niemoeller@diakonie-sw.de (U.N.); andreas.arnold@diakonie-sw.de (A.A.); josef.rosenbauer@diakonie-sw.de (J.R.); 2Department of Neurology, Justus-Liebig-University, 35392 Giessen, Germany; patrick.schramm@neuro.med.uni-giessen.de; 3Epidemiology, IQVIA, 60549 Frankfurt, Germany; karel.kostev@iqvia.com

**Keywords:** comprehensive geriatric care, older patients, EEG

## Abstract

Background: Comprehensive geriatric care (CGC) is a multidisciplinary approach developed to meet the needs of older patients. Electroencephalography (EEG) provides valuable information for monitoring the cerebral function. As a surrogate, EEG findings may help to estimate the course of diseases as well as the treatment outcomes. Objectives: Therefore, the aim of the present study is to investigate EEG findings in older patients receiving CGC. Methods: Patients with an initial EEG in a geriatric unit between May 2019 and April 2020 and treated using the CGC approach were analyzed. EEG abnormalities were defined as generalized (diffuse) background slowing and/or intermittent/persistent focal slowing and/or epileptiform discharges. Assessment results for the Barthel index (BI), Tinetti Balance and Gait test (TBGT), and Timed Up and Go test (TUG) were analyzed in relation to the presence of EEG abnormalities. Results: The study included 398 patients (mean age: 83.0 ± 6.57 years, 69.3% were female). Abnormal EEG patterns were found in 94 (23.6%) patients. Patients with EEG abnormalities had a mean age of 83.4 years versus a mean of 82.8 years in those without (*p* = 0.451). Based on all calculated scores, the majority of the patients improved after CGC, with a tendency to higher-grade improvements in those without EEG abnormalities (BI: 86.2% vs. 75.5%, *p* = 0.024; TUG: 53.3% vs. 31.9%, *p* < 0.001); for TBGT, only a gradual difference was detected (TBGT: 79.9% vs. 71.3%, *p* = 0.088). The presence of EEG abnormalities was associated with the parameters dementia (36.2% vs. 22.4%, *p* = 0.010), known epilepsy/seizure (19.1% vs. 5.9%, *p* < 0.001), structural brain lesion (47.9% vs. 19.7%, *p* < 0.001), and delirium (9.6% vs. 3.6%, *p* = 0.030) during hospitalization. Conclusions: We found EEG abnormalities in almost a quarter of the patients treated in the geriatric unit. In older patients, the presence of EEG abnormalities is associated with lower improvements after CGC.

## 1. Introduction

In recent decades, health care systems have been faced with an increasing number of older multimorbid patients, requiring continuous structural adjustment. Among other things, this has prompted the emergence of comprehensive geriatric care (CGC), a multidisciplinary intervention strategy developed to satisfy medical care requirements but also focusing on patients’ functional abilities in order to improve and/or maintain their independence [1,2,3]. Many investigations have indicated that patients might benefit from this approach [4,5]. However, there are a number of factors that may influence a patient’s functional abilities as well as their ability to recover [6,7]. In particular, disorders of the central nervous system such as ischemic or hemorrhagic stroke, tumors, epilepsy, encephalopathy, delirium, or dementia may cause deficits and affect rehabilitation [6,7]. Electroencephalography (EEG) is an established neurological procedure that provides valuable information for estimating and monitoring cerebral function and determines cognitive performances [8]. In this context, EEG abnormalities have been identified in many investigations as surrogates for the course of diseases as well as for treatment results [9,10]. In conclusion, EEG findings presumably could help to estimate performances in older patients scheduled for CGC. To the best of our knowledge, no data are available evaluating EEG findings in older patients undergoing CGC.

It is for this reason that we aimed to investigate EEG abnormalities in older patients treated in a geriatric unit and to analyze functional outcomes after CGC.

## 2. Materials and Methods

### 2.1. Patients

All patients who underwent CGC between May 2019 and April 2020 with the complete documentation of functional outcome assessments (BI, TBGT, and TUG) and EEG examination during hospitalization were selected for the analysis. As our study group represented a convenience sample, including all eligible consecutive patients, no specific exclusion criteria were applied. In addition, findings from further assessment procedures such as the Mini Mental Status Examination (MMSE), Shulman’s Clock Drawing test, and the Geriatric Depression Scale (GDS) conducted on admission were analyzed [11,12,13]. Demographic parameters, information on comorbidities, short-term adverse events, and EEG findings were extracted from clinical records.

### 2.2. Comprehensive Geriatric Care (CGC)

The procedure includes the assessment of older patients’ mobility and performance in coping with the basic activities of daily living and the evaluation of their cognitive and emotional status as well as their social environment. The standardized assessment includes BI, TBGT, and TUG prior to and after treatment and is supplemented by MMSE, GDS, and Shulman’s Clock Drawing Test on hospital admission [11,12,13,14,15]. Hospitalization for CGC is scheduled for a minimum of 2 weeks and consists of 20 treatment units, each 30 min long (physiotherapy/occupational therapy/speech therapy/psychological care). Based on the detailed evaluation of the patient’s medical needs and functional deficits, multidisciplinary teams develop personalized treatment strategies. Treatment progress is discussed weekly in a multidisciplinary team conference, and the individual treatment strategy is adapted to patients’ deficits.

### 2.3. Barthel Index (BI)

Patients’ ability to cope with basic activities of daily living (ADL) was expressed using the BI as a widely-used score covering 10 different items (dressing, walking, grooming, transfer, climbing stairs, using toilet, bathing, urinary and stool continence, and ingestion). BI values are shown on a scale from 0 to 100, and the higher the value, the better the ADL performance [14].

### 2.4. Tinetti Balance and Gait Test (TBGT)

The TBGT is used to assess balance and gait and indicates the individual fall risk of the older adult. Balance is evaluated in sitting and standing positions, when rising from and sitting down in a chair, while rotating through 360°, and after gentle pressure on the chest. Gait is assessed based on the length, height, symmetry, and continuity of the steps. Each element is assigned 0–2 points, and a maximum value of 28 can be achieved. Lower TBGT scores indicate a higher risk of falling [16].

### 2.5. Timed up and Go Test (TUG)

The TUG focuses on patient mobility. During the TUG, the time a patient needs to get up from a chair, walk three meters, turn, walk back, and sit down again is recorded [15]. Based on the time the patient requires to perform these actions, the TUG assigned the patient to one of five categories: (5) inability to walk; (4) >30 s for TUG completion, (3) 20–29 s for TUG completion, (2) 10–19 s for TUG completion, and (1) <10 s for TUG completion. Lower TUG scores indicate better mobility. 

### 2.6. Electroencephalography

EEG examinations were performed using the 10/20 system (Neurofax EEG 1200, Nihon Kohden Europe, Rosbach, Germany) and were carried out over the scheduled time of 20 min. Data sets were stored and evaluated in the intradepartmental archiving system (polaris.one, Nihon Kohden Europe, Rosbach, Germany). EEGs were recorded by a neurophysiological assistant and evaluated by a board-certified physician. EEG patterns were defined as abnormal when one of the following conditions was satisfied: (1) generalized (diffuse) background slowing in the theta (4–<8/s) or delta (0.1–<4/s) band, (2) focal slowing (intermittent or persistent background slowing of a focal brain region), and (3) epileptiform discharges (such as spikes, polyspikes, sharp waves, sharp-and-slow-wave complexes, and spike-and-slow-wave complexes).

### 2.7. Statistical Analyses

Data sets were described as mean ± standard deviation (SD) or median and interquartile range (IQR; quartile 1–quartile 3). Categorical variables are presented as percentages and frequencies. A normal distribution was proved by the Kolmogorov–Smirnov test. The Mann–Whitney U-test for unpaired samples and a sign test for paired samples were carried out for nonparametric data. Students *t*-test was performed in case of normal distribution. Comparing relative frequencies, the Fisher’s exact test was used. Statistical analysis was carried out with PSPP software; (version 1.4.1, GNU project).

### 2.8. Ethical Approval

For the current study, ethical approval from the local ethical committee was obtained (Ethikkommission der Ärztekammer Westfalen-Lippe und Westfälischen Wilhelms-Universität, protocol number 2021-175-f-S).

## 3. Results

In the presented retrospective study, 398 patients were included (mean age: 83.0 ± 6.57 years). Among them, 276 (69.3%) were female. Abnormal EEG patterns were found in 94 patients (23.6%). Generalized (diffuse) background slowing was detected in 57 (14.3%), focal slowing in 41 (10.3%), and epileptiform discharges in two (0.5%) patients. The proportion of patients with improvements for BI, TBGT, and TUG was higher in the subgroup of patients without EEG abnormalities (BI: 86.2% vs. 75.5%, *p* = 0.024; TBGT: 79.9% vs. 71.3%, *p* = 0.088; and TUG: 53.3% vs. 31.9%, *p* < 0.001) (Figure 1).

The diagnoses of dementia (36.2% vs. 22.4%, *p* = 0.010), epilepsy/seizure (19.1% vs. 5.9%, *p* < 0.001), structural brain lesion (47.9% vs. 19.7%, *p* < 0.001), and delirium (9.6% vs. 3.6%, *p* = 0.030) were more common in patients with EEG abnormalities (Table 1). On hospital admission, worse scores in BI, TBGT, TUG, MMSE, and Shulman’s clock-drawing test were detected in the subgroup of patients with EEG abnormalities; results are indicated in Table 1.

Comparing the outcome parameters prior versus after CGC, improvements were found in both groups. In those patients with EEG abnormalities, improvements for all three parameters were detected (BI: median 35 (IQR: 23.75–51.25) to 47.5 (IQR: 33.75–66.25) (*p* < 0.001); TUG: median 4.5 (IQR: 3.75–5) to 4 (IQR: 3–5) (*p* < 0.001); and TBGT: median 7 (IQR: 1.75–14) to 11 (IQR: 4–18) (*p* < 0.001)). In the subgroup of patients without abnormal EEG findings, BI increased from median 50 (IQR: 35–65) to 65 (IQR: 55–80) (*p* < 0.001), TBGT increased from 13 (IQR: 5–19) to 18 (IQR: 13–21.75) (*p* < 0.001), and TUG improved from median 4 (IQR: 3–5) to 3 (IQR: 2–4) (*p* < 0.001). The results are summarized in Table 2.

## 4. Discussion

In our study population of older patients treated in a geriatric unit, we found EEG abnormalities in 23.6% of cases. The presence of these abnormalities was associated with previously diagnosed dementia, epilepsy, or brain lesion. Patients with EEG abnormalities were more prone to suffer delirium during hospitalization (9.6% versus 3.6%). While EEG abnormalities did not appear to influence improvements in ADL tasks or gait and balance after CGC, they did have a negative influence on improvements in walking ability. Patients without EEG abnormalities performed better in the TUG test after CGC in 53.3% of cases, while such improvements were observed in only 31.9% of individuals with EEG abnormalities.

Compared to previous investigations, the frequency of 23.6% EEG abnormalities detected in our study seems to be low. Reviewing the literature, the occurrence of EEG abnormalities in older people ranges between 30% and >90%, depending on the characteristics of the population investigated and the EEG criteria for abnormalities [17,18,19,20,21]. Ter Schuur et al. found EEG abnormalities in 39% of older patients with mood disorders and in 41% of older patients with subjective cognitive decline [17], while a comprehensive investigation conducted by Liedorp et al. detected EEG abnormalities in 36–97% of subjects [18]. The frequency depended on the patient’s disorder and the definition of the abnormality; in patients with “subjective complaints” only, i.e., healthy older individuals, the calculated frequency of EEG abnormalities was 36% [18]. There are a number of reasons that could be responsible for the discrepancy in EEG abnormality rates detected in our study and those described in the literature. Our patients represent a heterogeneous group of elderly patients and were selected primarily for comprehensive geriatric care and not for proving EEG findings in a pre-specified group. Secondly, the conditions for the examination differ from the settings described in other studies; our EEG was performed as part of the clinical routine, while in other investigations the EEG was a step in a clinical study. It could be speculated that the study investigators were more focused on the identifying all potential EEG findings of interest. 

Our investigation depicted the number of elderly patients scheduled for CGC who presented abnormal findings in the routinely performed EEG. However, our results indicate that an abnormality detected in the EEG would have an unfavorable impact on improvements in walking ability in older patients selected for CGC. Interestingly, improvements in basic daily activities and gait and balance remain unaffected by the presence of abnormalities in the EEG. It is possible that procedural factors, when performing these assessments, determined our findings. Albeit both tests addresses have comparable functions, there are gradual differences. However, as there is no clear and conclusive explanation for this result, further examinations are needed to demonstrate the influence of EEG findings on the rehabilitation capability of older patients scheduled for comprehensive geriatric care.

Our study indicates that EEG abnormalities detected in EEGs prior to planned CGC could be regarded as useful surrogates, indicating potential favorable and negative impacts on the course of the procedure. The addition of such information when estimating the cognitive status prior to CGC in individuals who might have difficulties completing the cognitive assessment appears beneficial. It is well known that the presence of EEG abnormalities is associated with cognitive decline or dementia [22,23]. In this context, the association between EEG abnormalities and known dementia depicted in our study is in line with previous investigations [22,23]. This observation also matches our results as detected in the MMSE; patients with EEG abnormalities performed more poorly in the assessment than those without. We also observed that higher numbers of patients in the subgroup with EEG abnormalities suffered delirium in the course of the CGC. It can be speculated that an EEG abnormality detected prior to CGC could predict subsequent delirium. This useful information could potentially help to optimize the utilization of resources. However, additional studies are necessary to present further evidence of the relevance and the predictive value of these factors in elderly patients, especially in the context of planned CGC.

In our study, we observed an association between EEG abnormalities and previously diagnosed epilepsy or brain lesion. This finding is evident and expected and is already described in numerous previous investigations [24]. It helps to underline the reliability of our study and the related results.

The two major strengths of this study are the high number of patients treated using CGC with an initial EEG examination available for analysis, and detailed analyses using real-world data. This study is also subject to several limitations that should be acknowledged at this point. First, the lack of a control group within the area of regular subject-specific care for the purpose of comparing outcomes could be considered a major limitation of the present study. Second, since our study was conducted using clinical routine data, rigorous settings for proving therapy effects are missing. No causal relationships could therefore be reported due to the retrospective nature of the analysis; instead, we focused solely on associations.

## 5. Conclusions

We detected EEG abnormalities in almost a quarter of the patients treated in a geriatric unit when the procedure was performed prior to CGC. EEG abnormalities seem to be associated with a better outcome with respect to improvements in the basic activities of daily living and balance and gait but are a less favorable factor with regard to walking ability. In the context of planned CGC, EEG abnormalities could indicate cognitive decline and may potentially predict the development of delirium in the course of the procedure.

## Figures and Tables

**Figure 1 brainsci-12-00839-f001:**
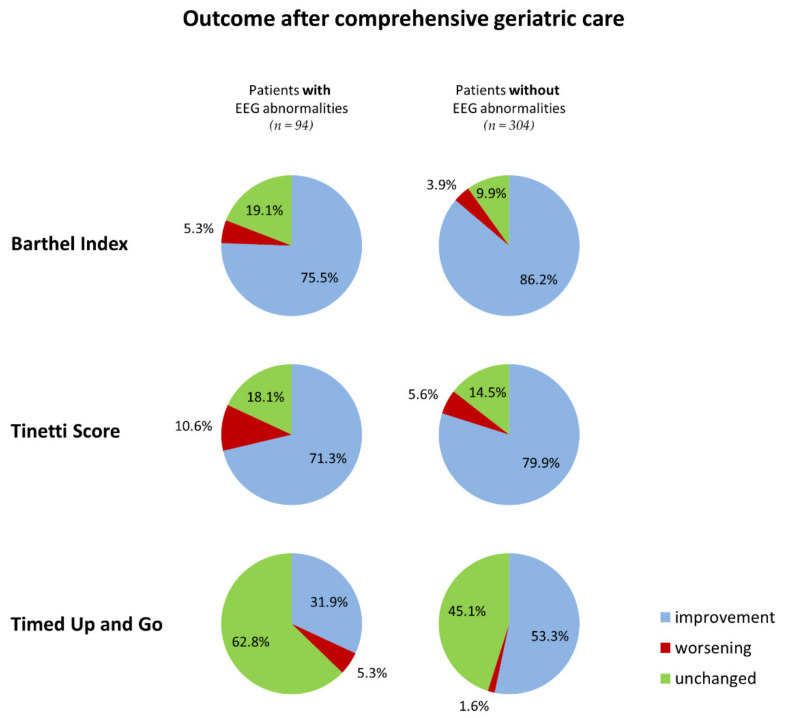
Functional outcome after comprehensive geriatric care in older adults with and without EEG abnormalities.

**Table 1 brainsci-12-00839-t001:** Factors associated with EEG abnormalities.

	Total Group (*n* = 398)	Patients with EEG Abnormalities (*n* = 94)	Patients without EEG Abnormalities (*n* = 304)	*p*-Value
**Age** (mean ± SD, years)	83.0 ± 6.57	83.4 ± 7.02	82.8 ± 6.43	0.451
**Sex**				
Female	276 (69.3%)	58 (61.7%)	218 (71.7%)	0.074
Male	122 (30.7%)	36 (38.3%)	86 (28.3%)	
**Comorbidities**				
Hypertension	320 (80.4%)	72 (76.6%)	248 (81.6%)	0.300
Diabetes mellitus	125 (31.4%)	31 (33.0%)	94 (30.9%)	0.705
Heart failure	109 (27.4%)	28 (29.8%)	81 (26.6%)	0.597
Coronary heart disease	117 (29.4%)	28 (29.8%)	89 (29.3%)	>0.999
Atrial fibrillation	132 (33.2%)	30 (31.9%)	102 (33.6%)	0.803
Carcinoma/Tumor	99 (24.9%)	18 (19.1%)	81 (26.6%)	0.172
Chronic obstructive pulmonary disease	44 (11.1%)	9 (9.6%)	35 (11.5%)	0.708
Asthma	10 (2.5%)	3 (3.2%)	7 (2.3%)	0.706
Dementia	102 (25.6%)	34 (36.2%)	68 (22.4%)	**0.010**
Depression	47 (11.8%)	12 (12.8%)	35 (11.5%)	0.717
Epilepsy/Seizure	36 (9.0%)	18 (19.1%)	18 (5.9%)	**<0.001**
Structural brain lesion ^#^	105 (26.4%)	45 (47.9%)	60 (19.7%)	**<0.001**
**Short-term adverse events while hospitalized**				
Delirium	20 (5.0%)	9 (9.6%)	11 (3.6%)	**0.030**
Pneumonia	14 (3.5%)	3 (3.2%)	11 (3.6%)	>0.999
Urinary tract infection	57 (14.3%)	18 (19.1%)	39 (12.8%)	0.132
Hypokalemia	139 (34.9%)	39 (41.5%)	100 (32.9%)	0.138
Hyperkalemia	32 (8.0%)	10 (10.6%)	22 (7.2%)	0.284
Hyponatremia	40 (10.1%)	12 (12.8%)	28 (9.2%)	0.329
Hypernatremia	23 (5.8%)	6 (6.4%)	17 (5.6%)	0.801
Hypocalcemia	182 (45.7%)	50 (53.2%)	132 (43.4%)	0.099
Hypercalcemia	12 (3.0%)	3 (3.2%)	9 (3.0%)	>0.999
**Functional assessments**				
Barthel index on admission *	45 (35–60)	35 (23.75–51.25)	50 (35–65)	**<0.001**
Tinetti on admission *	12 (3–18)	7 (1.75–14)	13 (5–19)	**<0.001**
Geriatric depression scale * (*n* = 354)	4 (2–6)	5 (2–6)	4 (2–6)	0.152
Geriatric depression scale ≤ 5	241 (68.1%)	47 (19.5%)	194 (80.5%)	0.482
Geriatric depression scale > 5	113 (31.9%)	26 (23.0%)	87 (77.0%)	
Timed up and go on admission *	4 (3–5)	4.5 (3.75–5)	4 (3–5)	**0.008**
MMSE * (*n* = 345)	26 (22–28)	24 (19–27)	26 (23–28)	**0.003**
Shulman’s clock-drawing test * (*n* = 276)	3 (2–4)	4 (3–4)	3 (2–4)	**0.001**

* Presented as median and interquartile ranges. ^#^ Includes acute and previous stroke/intracranial hemorrhage, intracranial tumor, and unspecified head injuries.

**Table 2 brainsci-12-00839-t002:** Barthel index, Tinetti score, and Timed Up and Go test; values for geriatric patients with and without EEG abnormalities prior to versus after comprehensive geriatric care (CGC).

	Prior to CGC	After CGC	*p*-Value
**Patients with EEG abnormalities**			
Barthel index (median, IQR)	35 (23.75–51.25)	47.5 (33.75–66.25)	**<0.001**
Tinetti score (median, IQR)	7 (1.75–14)	11 (4–18)	**<0.001**
Timed Up and Go test (median, IQR)	4.5 (3.75–5)	4 (3–5)	**<0.001**
**Patients without EEG abnormalities**			
Barthel index (median, IQR)	50 (35–65)	65 (55–80)	**<0.001**
Tinetti score (median, IQR)	13 (5–19)	18 (13–21.75)	**<0.001**
Timed Up and Go test (median, IQR)	4 (3–5)	3 (2–4)	**<0.001**

## Data Availability

Data sharing is possible on demand by contacting the corresponding author (christian.tanislav@diakonie-sw.de).

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
