# Peer review of "Electroencephalography Findings in Older Adults Undergoing Geriatric Treatment: A Surrogate for the Outcome?"

_brainsci, 2022, doi:10.3390/brainsci12070839_

Round 1

Reviewer 1 Report

Dear authors, I agree with the importance of comprehensive geriatric care (CGC) for the older adults and I think your goal was clear and relevant but your article needs substantial improvements to be published, mainly in the methods and discussion.

Here are my considerations:

1.Format your article according to Brain sciences

2.Abstract: you need to rewrite methods and results since I recommend to create different sample group. Conclusions: you cannot say EEG abnormalities are predictive, there are other variables that you did not considered in your study to be able to conclude that; Reduce your keywords. You have 7 keywords: “geriatric unit, EEG findings, older patients, comprehensive geriatric care, geriatrics, 28 functional outcome, electroencephalography. “. I recommend Older adults; comprehensive geriatric care; EEG;” . Why this is important : “To the best of our knowledge, no data are available evaluating the relevance of EEG findings with respect to the performance of older patients after CGC.” Justify in the text before you state this.

3.Introduction: It´s not clear enough and you need to improve and reinforce the impact of CGC and the other factors that affect the functional abilities. You just stated “However, there are a number of factors that may influence a patient’s functional abilities as well as their ability to recover (6,7). In particular, disorders of the central nervous system such as ischemic or hemorrhagic stroke, tumors, epilepsy, encephalopathy, delirium or dementia may cause deficits and affect rehabilitation.”

4.Methods: add your study design; add inclusion and exclusion criteria. You have to create a minimum of two groups, one with EEG abnormalities and other without abnormalities, and then you have to do statistics to present your results and discuss your results. You have to calculate your sample power (e.g. G*Power) and effect size. You need to compare before and after CGC in booth groups.

5.Results: It’s too confuse. Where is figure 1? You have to format the tables.

6.Discussion: The only result really discussed was the occurrence of EEG abnormalities. And the other results? You have to discuss BI, TBGT,TUG results also. Your discussion is to weak.

7.Conclusion: How you can conclude that balance improved but walking capacity decreased?? I not agree with this statement: “In the context of planned CGC, EEG abnormalities could indicate cognitive decline and may potentially predict the development of delirium in the course of the procedure. “ how you justify this conclusion?

Author Response

see uploaded document

Reviewer 2 Report

Electroencephalography Findings in Older Adults Undergoing Geriatric Treatment: A Surrogate for the Outcome?(brainsci-1751519)

This manuscript aims at determine the EEG characteristics in older adults undergoing geriatric treatment and the potential association between EEG abnormalities with treatment outcomes. The results reveled that EEG abnormalities existed in almost a quarter of the patients treated in the geriatric unit and EEG abnormalities seem to be predictive with some respects to the impact of CGC on the patient outcome. Overall, this topic is important. However, some concerns appeared after reading the whole manuscript.

1. “Patients with EEG abnormalities were older (mean: 83.4 years versus 82.8 years; p=0.451)”. The p value is larger than 0.05 and it is not appropriate to say older when the difference is not reach significance. It is the same case for “TBGT: 79.9% vs. 71.3%, p=0.088”. It should be careful when drawing conclusions.

2. More details about the Comorbiditiesshould be provided, such as depression, what is it cutoff point of Geriatric Depression Scale (GDS)?

3. EEG patterns were defined as abnormal when one of the following conditions was satisfied: 1) generalized EEG background activity <8 Hz, 2) focal slowing, 3) asymmetries, 4) epileptiform discharges.More details about the analysis of EEG data should be provided. What does “background activity”, “focal slowing”, “asymmetries” and “epileptiform discharges” mean and how to analysis them should be clearly stated to help the readers get better understanding of the current results.

4. What does “ADL” mean? The abbreviation should be explained when it firstly appeared in the context.

5. The language shold be edited by a native speaker.

Author Response

see aploaded document

Round 2

Reviewer 1 Report

Dear authors,

you have attended some of my suggestions.

But you must a) format your article according to Brain Sciences; b) add keywords in abstract; c) add exclusion criteria, if you do not have any exclusion criteria you must say that this was a convinience sample; 

Author Response

Please find revised manuscript attached.

Reviewer 2 Report

No further concerns.

Author Response

(The authors gave the same response as above.)
